

# Technical note: Phase space depiction of CCN activation and cloud droplet diffusional growth

Wojciech W. Grabowski[1] and Hanna Pawlowska[2]

[1]MMM Laboratory, NSF National Center for Atmospheric Research, Boulder, CO, USA
[2]Institute of Geophysics, Faculty of Physics, University of Warsaw, Warsaw, Poland

*Correspondence to*: Wojciech W. Grabowski (grabow@ucar.edu)

**Abstract.** A novel way to represent cloud condensation nuclei (CCN) activation and cloud droplet growth by the diffusion of water vapor is introduced. The key is to apply a phase space diagram that plots the radius of a liquid droplet (deliquesced CCN or cloud droplet) versus the difference between the ambient supersaturation and the equilibrium supersaturation corresponding
to the droplet radius. The latter combines the droplet and environmental characteristics, and it determines whether a droplet grows or evaporates. The diagram can be used to depict in a straightforward way key microphysical processes of CCN activation and deactivation, as well as a haze or cloud droplet transition from growth to evaporation. To show its utility, the diagram is applied to an idealized simulation of CCN activation and cloud droplet growth inside a rising turbulent air parcel and to simulations of microphysical processes inside a laboratory apparatus, the Pi cloud chamber. The adiabatic parcel mimics
microphysical processes near the base of a natural cumulus or stratocumulus cloud. The Pi chamber simulations represent microphysical transformations in moist turbulent Rayleigh-Benard convection with CCN proceeding through cycles of activation, growth, evaporation, and deactivation. A more general version of the phase diagram that is independent of the CCN dry radius is also developed. The phase diagram allows simple interpretations of key microphysical processes and highlights differences between droplet formation in natural and laboratory clouds.

## 1 Introduction

Formation and growth of cloud droplets are keys to understanding properties of warm ice-free clouds. Recent developments in numerical cloud modeling, especially the growing interest in the Lagrangian particle-based microphysics (Shima et al. 2009; Arabas and Shima 2013; Arabas et al. 2015; Grabowski et al. 2019 and references therein; Dziekan et al. 2021; Hoffmann and Feingold 2021; Chandrakar et al. 2022, Yang et al. 2023; Morrison et al. 2024, among others) inspired us to revisit the problem
of cloud droplets formation and their diffusional growth in early stages of cloud development. The narrative below provides a textbook introduction to the problem, but we feel it is appropriate to revisit elementary concepts before moving to a specific tool we introduce in this manuscript.



Atmospheric cloud droplets form by a heterogeneous nucleation of microscopic aerosol particles that serve as catalysts

facilitating droplet formation. Those soluble aerosols are referred to as cloud condensation nuclei (CCN). Without CCN, it is impossible to form a water droplet directly from water vapor in the Earth atmosphere. The equilibrium water vapor pressure over a spherical droplet containing a soluble material is described by the Koehler equation. It was formulated close to a century ago by a Swedish meteorologist Hilding Koehler (Koehler 1936). The equation incorporates two factors: an increase of the saturation vapor pressure related to the surface tension of a curved droplet surface, and a reduction of the saturation pressure

because of the dissolved CCN. The resulting form of the Koehler curve is shown in Figure 1 and in its simplest form is given by the following equation:

$$S_{eq} = \frac{A(T)}{r} - \frac{\kappa r_d^3}{r^3} \qquad (1)$$

where $S_{eq}$ is the equilibrium supersaturation over a spherical droplet with radius $r$, $r_d$ is the CCN dry radius that describes the amount of solute inside the droplet, $\kappa$ is the hygroscopicity parameter, and $A$ is a coefficient that weakly depends on the temperature [$A(273\ K) = 0.0012\ \mu m$]. Here we use a notation introduced by Petters and Kreidenweis (2007) that allows for characterization of CCN by a single parameter $\kappa$. However, numerical simulations discussed in this paper include a more general formulation of the droplet growth equation that includes kinetic, solute, and surface tension effects, see Grabowski et

al. (2011; Eqs. 2, 3, 10, and 11 therein).

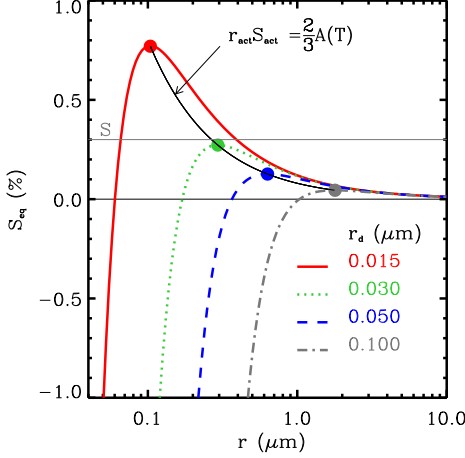

Fig. 1. Examples of Koehler curves for selected dry radii of NaCl ($\kappa = 1.28$) and temperature of 273.16 K. Critical supersaturations and critical radii are marked as large dots. See text for discussion.


Koehler curves shown in Fig. 1 feature maxima that define the so called critical or activation parameters: the activation radius $r_{act}$, and the activation supersaturation $S_{act}$. Exact values of these parameters depend weakly on the temperature, on the



hygroscopicity parameter ($r_{act} \propto \sqrt{\kappa}$, $S_{act} \propto 1/\sqrt{\kappa}$), and on the dry CCN radius ($r_{act} \propto r_d^{3/2}$, $S_{act} \propto r_d^{-3/2}$). Moreover, it follows from (1) that the critical radius and critical supersaturation are inversely proportional to each other because $r_{act} S_{act} =$
$\frac{2}{3} A(T)$ as shown in the figure. For the formation and early growth of cloud droplets, the most important feature of the Koehler curve is its dependence on the dry CCN radius. The peak of the Koehler curve divides nonactivated droplets ($r < r_{act}$), also called haze droplets, from the activated droplets referred to as cloud droplets ($r > r_{act}$). The growth rate of haze and cloud droplets is proportional to $S - S_{eq}$. For a constant $S$ smaller than the activation supersaturation (i.e., as $S$ for the smallest $r_d$ in Fig. 1), haze droplets are in a stable equilibrium. This is because a change of the environmental supersaturation (i.e., increase
or decrease of $S$) leads to an adjustment of the haze droplet radius (increase or decrease, respectively) towards the new equilibrium size. Activated droplets (i.e., cloud droplets) are in an unstable equilibrium with the environmental supersaturation $S$, that is, they continue to grow away from $r_{act}$ as long as $S > S_{eq}$, or evaporate towards $r_{act}$ when $S < S_{eq}$. In the latter case, they may deactivate (i.e., their radius may become smaller that the critical radius) and continue evaporating until $S = S_{eq}$. Another important feature of the Koehler curve is that for sufficiently large droplets (say, with radii larger than about 10 $\mu$m)
the equilibrium supersaturation converges to zero, that is, to the conditions over a plane pure water surface.

Supersaturated condition leading to the cloud droplet formation can originate from two different processes: i) uniform cooling of an air parcel (e.g., by radiative processes or by parcel expansion), or ii) mixing of two undersaturated air parcels with different thermodynamic conditions. Both these situations will be considered in this paper. For convective clouds, the first
mechanism involves adiabatic expansion of the air parcel carried by the sub-cloud updraft towards the cloud base. A common approach to study the cloud base CCN activation in natural clouds is to consider adiabatic parcel rising through the cloud base (e.g., Pruppacher and Klett 1997; Reutter et al. 2009; Ghan et al. 2011). The second mechanism comes from the curvature of the saturated water vapor pressure temperature dependence. As a result, isobaric mixing of two undersaturated air parcels can lead to supersaturated conditions. This mechanism is used in the laboratory cloud chamber at the Michigan Technological
University referred to as the Pi chamber (e.g., Chang et al. 2016; Chandrakar et al. 2016; see http://phy.sites.mtu.edu/cloudchamber/). Air flow inside the Pi chamber is driven by the temperature and humidity differences between 1-m-apart horizontal lower and upper boundaries. Those differences drive moist turbulent Rayleigh–Bénard convection within the chamber. The supersaturation inside the chamber comes from the isobaric turbulent mixing of moist warm air rising from near the lower boundary with colder air descending from near the upper boundary.

Once air becomes supersaturated, CCN activation and formation of cloud droplets takes place. First droplets form on the largest CCN because their critical supersaturation is the lowest (see Fig. 1). However, large CCN typically lag the environmental relative humidity increase and their activation may take place after activation of smaller CCN. The largest CCN (with dry radii larger than 1 micron, referred to as giant CCN) may never reach their critical radius, but they behave like regular droplets.
With the further supersaturation increase, smaller CCN can get activated and additional cloud droplets can form. With a



sufficiently large number of growing cloud droplets, the supersaturation starts to decrease because of the activated droplets condensational growth combined with the temperature increase due to latent heating. With the supersaturation decrease, it is possible that some already activated droplets can evaporate and become deactivated (e.g., Arabas and Shima 2017; Grabowski and Pawlowska 2023). The smallest CCN have no chance to become cloud droplets because their critical supersaturation is

too high to be reached in natural clouds. They remain as haze droplets throughout the cloud lifecycle. It follows that details of the CCN distribution (total CCN concentration and the concentration as a function of the dry CCN radius) as well as the environmental conditions (primarily the cooling rate, but also the air temperature) determine the partitioning between those CCN that become cloud droplets and those that remain as haze droplets.

This paper presents a novel way to represent CCN activation and droplet growth that can be used to depict cloud base activation in natural clouds and microphysical processes within the Pi chamber. The key is to apply a phase space diagram that is introduced in the next section. The diagram can be used to show instantaneous partitioning of the cloud condensate into nonactivated (haze) and activated droplets, and – perhaps more interestingly – it can be used to show phase space trajectories of CCN activation and deactivation, as well as haze droplet or cloud droplet transition from growth to evaporation. The diagram

is applied in sections 3 and 4 to a simulation of CCN activation and initial droplets growth inside an adiabatic parcel filled with isotropic homogeneous turbulence discussed in Grabowski et al. (2022b), and to simulations of droplet formation and growth in the Pi chamber as examined in Grabowski et al. (2024). A version of the diagram that is independent of the dry CCN radius is introduced in section 5. A brief summary in section 6 concludes the paper.

**2 The phase space diagram**

Figure 2 shows the diagram that depicts key elements of CCN activation and liquid droplet growth by the diffusion of water vapor. On the diagram, the radius of the droplet is plotted as a function of the difference between the local supersaturation $S$ and the Koehler theory droplet equilibrium supersaturation $S_{eq}$. Because the equilibrium supersaturation $S_{eq}$ depends on the

droplet radius and physicochemical properties of the dissolved CCN, the horizontal axis on the diagram combines the environmental conditions with the properties of a haze (deliquesced) droplet or a cloud droplet. Since the droplet diffusional growth rate is proportional to the $S$ and $S_{eq}$ difference, the solid vertical line at $S - S_{eq} = 0$ separates droplets that grow when the difference is positive (the right half of the diagram) from those that evaporate on the left half because of the negative $S - S_{eq}$ difference. For small cloud droplets, the positive equilibrium supersaturation $S_{eq}$ can allow their evaporation even if the

ambient supersaturation is positive (e.g., Arabas and Shima 2017; Abade et al. 2018; Grabowski and Pawlowska 2023). For droplets with radius larger than several microns, $S_{eq}$ is close to zero and the $S - S_{eq}$ difference is close to the ambient supersaturation $S$. The solid horizontal line shows the activation (critical) droplet radius that separates activated CCN (i.e., cloud droplets) above the line from haze droplets (i.e., deliquesced CCN) below the line. Except for the different symbols and switched vertical and horizontal axes, the diagram in Fig. 2 is similar to that used in Arabas and Shima (2017, see Fig. 1 there).



However, as shown later in the paper, we focus on more general aspects of the CCN activation and droplets growth than Arabas and Shima (2017).

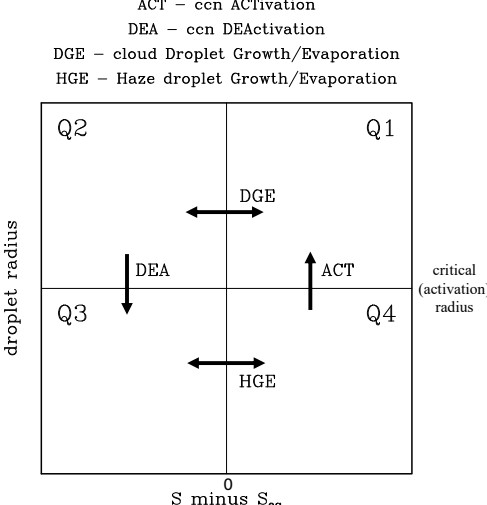

Fig. 2. The phase space diagram of CCN activation and droplet growth. The solid horizontal and vertical lines show the critical (activation) radius and the zero of the $S$ and $S_{eq}$ difference, respectively. The two lines divide the phase space into four quarters, Q1 to Q4, as marked in the figure. ACT, DGE, DEA, and HGE – defined above the panel – represent haze and cloud droplet transitions during growth and evaporation. See text for details.

The two solid lines divide the diagram into four quarters, Q1 to Q4 as marked in Fig. 2. The upper right quarter Q1 includes droplets with radius larger than the critical radius that feature positive $S$ - $S_{eq}$, that is, growing cloud droplets. The upper left quarter Q2 contains droplets larger than the critical radius and negative $S$ - $S_{eq}$, that is, evaporating cloud droplets. The lower two quarters include haze droplets that either grow towards the critical radius in the right quarter Q4 or evaporate in the left quarter Q3. The arrows across the solid lines in Fig. 2 represent processes that take place during CCN activation, droplet

growth and evaporation, and CCN deactivation. The activation (ACTivation) brings a deliquesced CCN (i.e., a haze droplet) across the critical radius line for a positive $S$ - $S_{eq}$ (i.e., from Q4 to Q1), whereas deactivation (DEActivation) brings an evaporating droplet across the same line in the left part of the diagram (i.e., from Q2 to Q3). ACT and DEA are one-way processes, that is, ACT and DEA only take place in the directions shown by the arrows. Horizontal transitions represent a change from a droplet growing to evaporating or evaporating to growing in the upper part of the diagram (marked as DGE,

Droplet Growth/Evaporation) and a change from a haze droplet evaporating to growing or growing to evaporating in the lower part; the latter is marked as HGE (Haze droplet Growth/Evaporation). In contrast to one-way ACT and DEA transitions, DGE and HGE can work both ways, that is, from left to right or from right to left depending on the $S$ - $S_{eq}$.



The next two sections discuss application of the phase diagram to CCN activation and droplet growth within an adiabatic
parcel filled with turbulence rising across the cloud base (Grabowski et al. 2022b) and to microphysical processes within the
simulated Pi chamber (Grabowski et al. 2024). The two sections document advantages of using the phase diagram to investigate
microphysical processes already discussed in those previous studies. In addition, the analysis further documents key
differences between the two environments that have already been discussed in Grabowski et al. (2024) and provides new
insights into microphysical processes in the simulated Pi chamber.

## 3 Application to CCN activation and droplet growth within a turbulent adiabatic parcel

Grabowski et al. (2022b; GTK22 hereafter) discuss impact of turbulence on CCN activation within an adiabatic parcel rising
across the cloud base. Activation within a nonturbulent parcel, a traditional methodology to study CCN activation at the cloud
base, provides a reference. To include the effects of turbulence on CCN activation, Grabowski et al. (2022a,b) introduced a
computational framework of a rising triply periodic computational domain carrying deliquesced CCN. The turbulent parcel
rises across the cloud base with a prescribed mean ascent rate that provides a spatially-uniform adiabatic cooling within the
parcel. Turbulent vertical velocity fluctuations provide small-scale inhomogeneities of the adiabatic cooling. The mean ascent
and turbulent fluctuations lead to CCN activation within the parcel. The turbulence is assumed to be isotropic and
homogeneous, and it is forced to maintain the assumed level of the turbulent kinetic energy within the parcel. The simulations
apply a $64^3$ m$^3$ computational domain and a 1 m grid length. The domain is filled with a weak-intensity turbulence (eddy
dissipation rate of 10 cm$^2$ s$^{-3}$ and rms turbulent velocity of about 0.2 m s$^{-1}$) that is arguably appropriate for cloud base turbulence
of natural clouds (e.g., Siebert et al. 2006; Borque et al. 2016).

The initial conditions within the domain feature a uniform 97% relative humidity (RH) across the volume with deliquesced
CCN at equilibrium with the ambient conditions. The 200 m parcel ascent across the cloud base leads to CCN activation
followed by the growth of activated cloud droplets. The CCN spectrum is based on field observations and contrasts cloud
droplet formation in either pristine or polluted CCN conditions. Overall, for larger mean ascent rates, a smaller dry CCN radius
separates activated and nonactivated CCN. As a result, for parcels with and without turbulence, cloud droplet concentration is
higher for a parcel rising faster across the cloud base. The key difference between nonturbulent and turbulent activation is a
blurriness of the separation between dry CCN sizes featuring activated and nonactivated (haze) CCN for the turbulent case,
especially for weak mean ascent rates. The blurriness comes from small-scale supersaturation fluctuations that allow CCN
activation and deactivation, instead of a sharp separation between cloud and haze droplets with no turbulence-driven
supersaturation fluctuations. This leads to significantly larger spectral widths in turbulent parcel simulations compared to the
nonturbulent parcel when activation is completed. The blurred transition from nonactivated to activated CCN leads to the CCN



activation fraction (i.e., the ratio between activated and total CCN for a given dry CCN radius) gradually increasing from zero for those CCN radii that never get activated to unity for those CCN radii that all get activated (see Fig. 10 in GTK22).

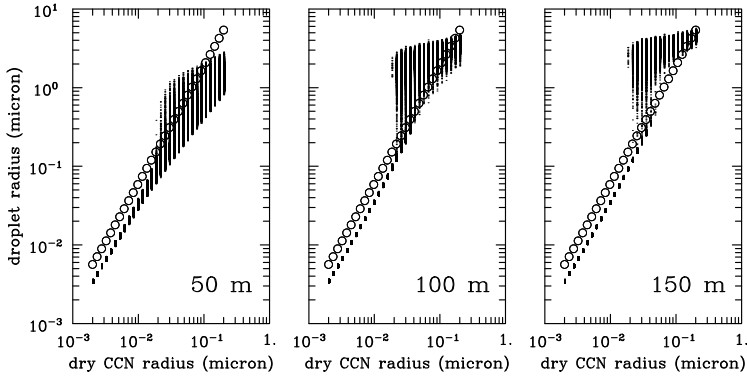


Fig. 3. Scatterplot of the droplet radius versus dry CCN radius at height of 50/100/150 m (left/middle/right panels) for the polluted simulation with 1 m s$^{-1}$ ascent rate from GTK22. Circles represent critical (activation) radius for each CCN bin. Only 5% out of about 7.5 million droplets used in the simulation is shown.

We use one of the GTK22 simulations to show how turbulent cloud base activation is represented in the diagram introduced in the previous section. Figure 3 shows scatterplots of the droplet radius versus dry CCN radius at three heights of the rising parcel: 50, 100, and 150 m for a simulation with polluted CCN distribution (total CCN concentration of 2,000 cm$^{-3}$) and mean parcel ascent rate of 1 m s$^{-1}$. A parcel height of 50 m is about the level at which the mean RH within the parcel reaches saturation. However, because of turbulent supersaturation fluctuations (with the standard deviation reaching close to 0.5%, see

Fig. 7 in GTK22), there are already some activated cloud droplets as shown in the left panel in Fig. 3. At 100 m – the middle panel in Fig. 3 – the number of activated droplets is larger because there are more points above the critical radius. The parcel height of 100 m is about 20 m above the level of the maximum mean cloud base supersaturation of about a couple tenths of 1% (see Fig. 7 in GTK22). Some of the largest CCN are still not activated despite having very low critical supersaturation. This is because those large CCN grow slowly and lag ambient RH in contrast to small CCN that respond rapidly to RH changes

prior to activation. At 150 m – the right panel – activated droplets continue to grow, small CCN remain as haze droplets, and only some of the largest dry CCN are still smaller than the critical (activation) radius.

Figure 4 uses the phase diagram introduced in the previous section to show droplets at 50, 100, and 150 m parcel height for two dry CCN radii, 30 and 100 nm. The results are for the turbulent parcel with 1 m s$^{-1}$ mean ascent rate as in Fig. 3. For the

nonturbulent parcel, all 30 and 100 nm CCN get activated. For the turbulent parcel, all 100 nm CCN get activated as shown in the lower right panel in Fig. 4. For 30 nm dry CCN within a turbulent parcel, about 82% get activated, and about 18% remain



nonactivated at 150 m parcel height. About one third of those nonactivated, 6% of the total, get activated and subsequently deactivated in the fluctuating supersaturation field. Upper panels in Fig. 4 show gradual CCN activation and separation between activated and nonactivated 30 nm CCN. Lower panels show a gradual approach of 100 nm CCN towards activation and

subsequent growth of cloud droplets.

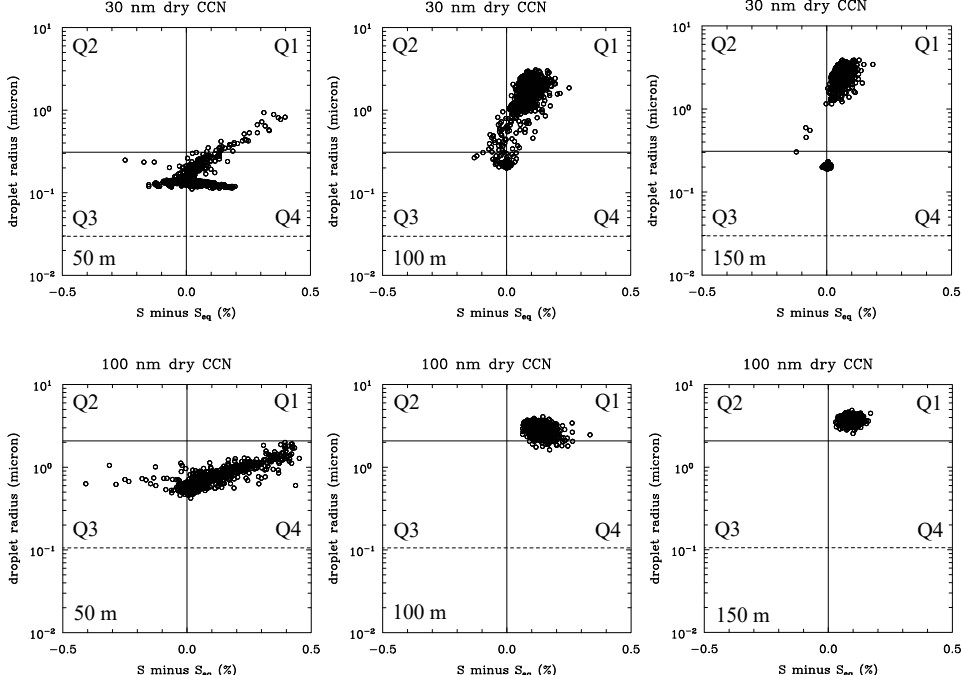

Fig. 4. The phase space diagram of CCN activation and droplet growth within an adiabatic turbulence-filled parcel rising across

the cloud base with the mean ascent of 1 m s$^{-1}$. The dashed horizontal line shows dry CCN radius. Left/middle/right panels are for the parcel at 50/100/150 m above the starting level. Lower/upper panels are for droplets with a dry CCN radius of 30/100 nm. Only about 0.3% of droplets are shown in each panel for clarity.

Figure 4 shows droplets at a particular time during parcel rise. However, the phase diagram is designed to show evolution of

selected droplets through cycles of droplet growth and evaporation. To highlight this aspect, Fig. 5 shows evolutions of a few randomly selected droplets formed on the 30 nm CCN dry radius using the same results as in Figs. 3 and 4. The left panel shows a traditional way to document parcel rise, with droplet radius plotted as a function of the parcel height. The colored lines show examples of the simulated droplet evolutions. Black lines show evolutions of five randomly selected 30 nm dry CCN that get activated and eventually form growing cloud droplets. Red lines show examples of five CCN that activate and

then deactivate after a couple tens of meters of the turbulent parcel rise. Finally, green lines show examples of CCN that stay



as deliquesced (haze) droplets and never reach the critical radius. As shown in Grabowski and Pawlowska (2023), activation and subsequent deactivation can also happen for a nonturbulent adiabatic parcel, especially in weak-updraft polluted clouds.

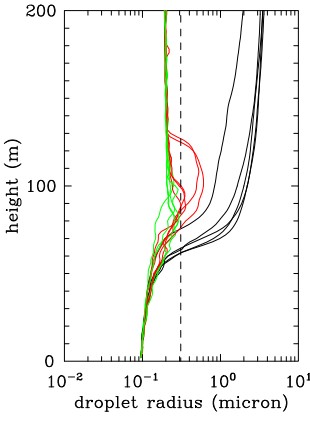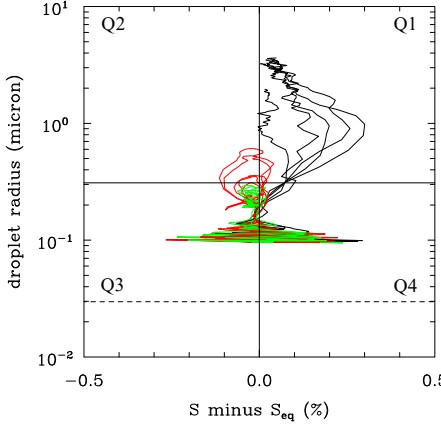


Fig. 5. Left panel: Droplet radius as a function of height for a set of droplets from simulation as in Figs. 3 and 4 and for dry CCN radius of 30 nm. Dashed vertical line shows the critical (activation) radius for the 30 nm dry CCN radius. Black color shows radii of 5 cloud droplets that continue growing after activation. Red color is for 5 CCN that get activated and
subsequently deactivated. Green color is for CCN that never get activated. Right panel: The phase space diagram for the same droplet set as in the left panel.

The right panel in Fig. 5 shows the same 15 droplets on the phase space diagram applying the same colors as in the left panel. The lines are not smooth because the simulation data are only saved every two seconds and ambient conditions change rapidly
in the turbulent environment. This is especially true during the early phase of the parcel rise, with deliquesced 30 nm dry CCN responding rapidly to turbulent fluctuations with haze droplet radii between 0.1 and 0.2 micron. Activated droplets (black lines) grow up to a few microns, and the same droplets can be identified between left and right panels. As the supersaturation decreases when the parcel moves away from the cloud base, the droplet trajectories approach the vertical line marking the zero of the $S - S_{eq}$. One of the black lines crosses the $S - S_{eq} = 0$ line a couple times because of the ambient supersaturation
fluctuations. This reverses the droplet growth for a short period of time that is not easily identified in the left panel. Red lines show CCN that activate and deactivate after reaching a few tenths of 1 micron radius. Green lines – not well visible in the right panel – show deliquesced CCN that occupy mostly quarter Q3 on the phase diagram. Both red and green lines approach equilibrium radius of about 0.02 micron as seen in the left panel.

**4 Application to the Pi chamber simulation**



Simulations from Grabowski et al. (2024; hereafter GKY24) are used to contrast activation in the adiabatic turbulent parcel from the previous section with simulated microphysical processes within the Pi chamber. Laboratory studies of moist Rayleigh-Benard convection within the Pi chamber focus on microphysical processes of droplet formation and growth in a turbulent environment. The chamber in its cuboid configuration has extent of 2 m by 2 m in the horizontal and 1 m in the vertical.

GYK24's Pi chamber simulations follow Grabowski (2020). Following the setup of one of laboratory experiments (e.g., Thomas et al. 2019), the lower and upper boundary temperatures are 299 K and 280 K, respectively, that is, a 19-K temperature difference. The side wall temperature is assumed to be the mean of the lower- and upper-boundary temperatures at 289.5 K. The computational domain is covered with a $51^3$ uniform grid, and horizontal and vertical grid lengths are 0.04 and 0.02 m, respectively. Lagrangian particle-based microphysics is used to represent growth of haze CCN and cloud droplets that include

kinetic, solute, and surface tension effects (Grabowski et al. 2011). For illustration, the phase diagram is applied to one of the simulations discussed in GYK24. The simulation, referred to as C40, applies an observationally-based dry CCN distribution with two lognormal modes centered at 20- and 75-nm radii, and the total CCN concentration of 40 cm$^{-3}$. The simulated mean droplet concentration for the C40 case is about 30 cm$^{-3}$ (see Fig. 3 in GYK24).


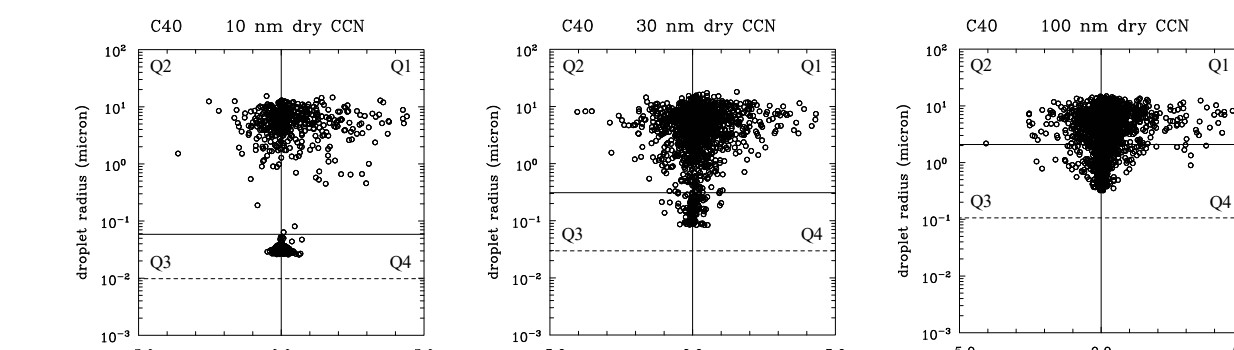

Fig. 6. The instantaneous phase space diagram for minute 20 of the C40 simulation from GKY24. Left/middle/and right columns are for dry CCN radius of 10/30/100 nm. Dash line show the dry CCN radius. Only 2% of all data points is used in

each panel.

Figure 6 shows an instantaneous phase space diagram for the dry CCN of 10, 30 and 100 nm. For clarity, the figure includes only 2% of all droplets from the last snapshots of model data, at minute 20 of the simulations. Droplets cover all four quarters, Q1 to Q4, with haze droplets and cloud droplets growing and evaporating at the time of the plot. Similar plots at different times

are similar, with the pattern shown in the figure similar across the entire computational domain.





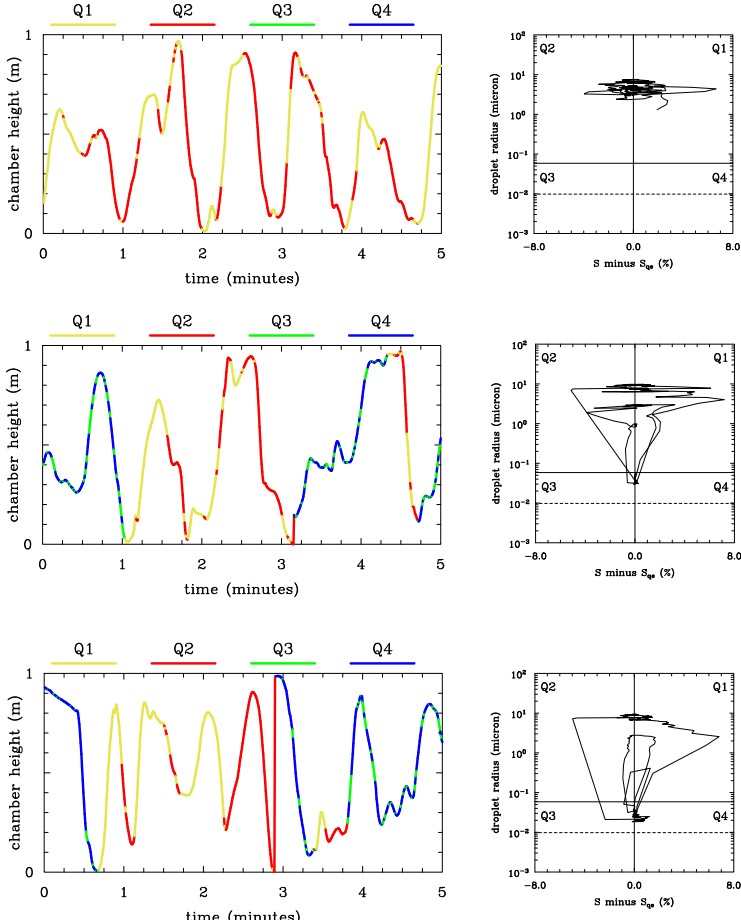

Fig. 7. Results for C40 simulations from GKY24 for 10 nm dry CCN radius. Left panels: Evolutions of droplet height in the
Pi chamber of randomly selected 3 droplets with colors marking droplet position on the phase space diagram. Colors defining
the quarters on the diagram (Q1 to Q4) are shown above each left panel. Right panels: Representation of droplet trajectories
on the phase diagram.

Figure 6 provides a static picture, that is, it does not include evolution of individual droplets. To show the evolutions, the C40
simulation was restarted at minute 20 and run for an additional 5 minutes. Evolutions for droplets with 10, 30, and 100 nm dry
CCN radius were saved every half second together with the $S - S_{eq}$ averaged over the preceding half second. The latter is
because the droplet radius change from one half second to another comes from the $S - S_{eq}$ averaged over the half second. The
idea is to show how individual droplets within the Pi chamber progress through their cycles of deliquescence, activation,
growth, evaporation, and deactivation while they move within the chamber carried by the chamber-scale flow. Figure 7 shows
example of model results for three randomly selected 10 nm dry CCN radius droplets. The left panels show evolutions of the





droplet vertical position within 1 m deep chamber together with the droplet trajectory on the phase diagram (right panels). The line color in the left panels shows the quarter in the right panel that the droplet is located at a given moment of time. Colors and corresponding quarters are shown above the left panels. Q1, Q2, Q3, and Q4 refer to, respectively, upper right, upper left, lower left, and lower right quarters as in Fig. 2. The plots are for three randomly selected droplets out of about 125,000 droplets

with a 10 nm CCN dry radius. As expected, the left panels show that droplets circulate up and down inside the chamber. However, as already mentioned in GKY24, the vertical droplet motion has little to do with droplet growth because the supersaturation comes from the turbulent mixing inside the chamber, and not from the vertical air motion and adiabatic cooling as in natural clouds.

As the panels in Fig. 7 show, two out of the three droplets go through cycles of droplet activation and deactivation (i.e., crossing the horizontal solid line) as well as growing and evaporating as haze droplets and cloud droplets (i.e., deliquesced CCN with radius smaller and larger, respectively, than the critical radius). The droplet depicted in the top row of Fig. 7 remains activated over the five minutes period, that is, it remains above the solid line in the right panels and its space trajectory only includes red and yellow colors.


One can count the time all 10 nm dry CCN radius droplets spend in each quarter of the diagram. The results are as follows: about 28% in Q1 (activated cloud droplets that grow); about 24% in Q2 (activated cloud droplets that evaporate); about 13% in Q3 (evaporating haze droplets); and about 35% in Q4 (growing haze droplets). The partitioning between activated droplets (Q1 and Q2) and haze droplets (Q3 and Q4) approximately agrees with Fig. 9 from GKY24 that shows about 50% activation

fraction for the CCN with 10 nm dry radius. The same statistics for 30 and 100 nm dry CCN show slightly higher numbers for Q1 and Q2 (consistent with activation fraction close to 60% for those CCN as shown in Fig. 9 from GKY24) and corresponding reductions of Q3 and Q4 (not shown). The lowest number for all three CCN is for Q3, 13%, 14%, and 20% for 10, 30, and 100 nm dry radius, respectively.

One can calculate transfer statistics between various quarters as shown in Fig. 2. Table 1 presents the average number of transfers per droplet during the 5 minute simulation for haze and cloud droplets formed on 10, 30, and 100 nm dry CCN in the C40 simulation. The statistics are for activation (ACT), deactivation (DEA), from growth to evaporation for cloud droplets (DGE) and haze droplets (HGE). Growth and evaporation for cloud droplets (between Q1 and Q2) and haze droplets (between Q3 and Q4) are separated into those from left to right (i.e., evaporation into growth) marked with the "+" sign, and those from

right to left (i.e., growth into evaporation) marked with the "-" sign. As the table shows, the transfer statistics are similar for the three dry CCN radii. The differences do not seem statistically significant considering the standard deviations among all droplets that are shown in the parentheses. The activation and deactivation numbers are small considering other transfer numbers in the table. This seems to agree with the results shown in Fig. 7 that documents frequent transfers between Q1 and Q2 as well as between Q3 and Q4, with smaller numbers of Q4 to Q1 and Q2 to Q3 transfers (i.e., ACT and DEA). The haze



droplet transfers (HGE) are the most frequent, and their number decreases with the dry CCN increase. The latter is likely because of an increased "inertia" of large CCN that follow ambient supersaturation fluctuations with more difficulty than small CCN.

Table 1. Averaged number of transfers per dry CCN defined in Fig. 2 for cloud and haze droplets formed on 10, 30, and 100
nm dry CCN radii during the 5 minutes of the additional simulation. Two-way transfers DGE and HGE are separated into those from left to right (+) and from right to left (-). The numbers in parenthesis show the standard deviation among all droplets for a given dry CCN radius.

|  | 10 nm | 30 nm | 100 nm |
| --- | --- | --- | --- |
| ACT | 3 (2) | 4 (2) | 3 (2) |
| DEA | 3 (2) | 4 (2) | 3 (2) |
| DGE+ | 12 (7) | 15 (7) | 13 (7) |
| DGE- | 14 (7) | 17 (6) | 16 (7) |
| HGE+ | 55 (31) | 46 (27) | 39 (25) |
| HGE- | 53 (31) | 44 (27) | 37 (25) |

**5 Phase space diagram independent of dry CCN radius**

The diagram introduced in section 2 and applied to simulations in sections 3 and 4 depends on a specific dry CCN radius. A more general phase diagram that can be applied to different CCN dry radii might also be of interest. One option might be to plot not the droplet radius on the vertical axis but the difference between the droplet radius and the activation (critical) radius. Such a modification makes the diagram independent of the dry CCN radius. However, in such a case, it is impossible to use
logarithmic scale on the vertical axis when the difference is negative for the haze droplets. The logarithmic scale is convenient because of a large radius range between haze and cloud droplets, and for that reason it was used in section 3 and 4 figures. Another option is to consider the ratio between the droplet radius and the activation (critical) radius that facilitates in a natural way application of the logarithmic scale. Fig. 8 shows the two options in the left and middle panels for droplets formed on three dry CCN radii, 10 nm (red color), 30 nm (green color) and 100 nm (blue color) from the C40 simulation discussed in the
previous section (e.g., Fig. 7). As shown in the left panel, the linear scale for the difference between the droplet radius and the critical radius is good for cloud droplets in the upper half of the diagram because they all have similar radius ranges. However, such a choice is poor for haze droplets because the critical radius varies between 60 nm for 10 nm dry CCN radius and about 2 microns for 100 nm dry CCN radius. In contrast, as shown in the middle panel in Fig. 8, using the ratio between droplet radius and the critical radius fits better the haze droplets, but then cloud droplets are separated as the ratio between the droplet radius and the critical radius vary over two orders of magnitude. A sensible compromise is to use the difference between
droplet radius and the activation (critical) radius for cloud droplets, and the ratio between droplet radius and the critical radius



for the haze droplets. Such an approach, shown in the right panel in Fig. 8, combines quarters Q1 and Q2 from the left panel and quarters Q3 and Q4 from the middle panel.

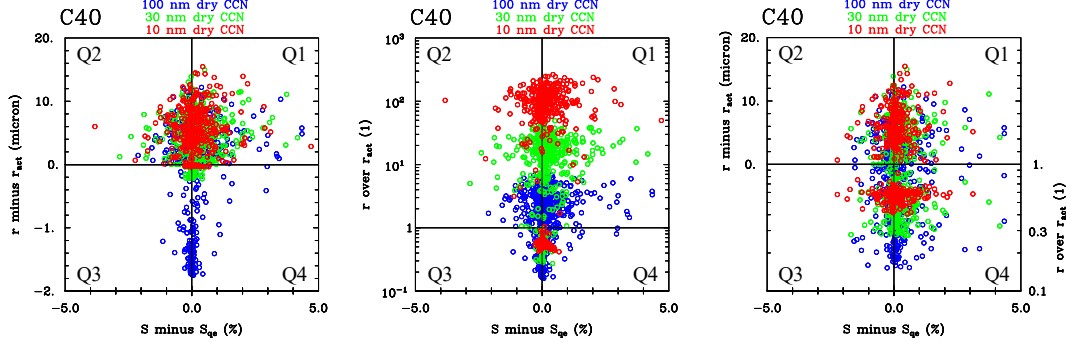


Fig. 8. Results for C40 simulations from GKY24. The same data as in Fig. 6, but on the modified phase diagram and with smaller number of points (0.5% for each dry CCN radius). Red/green/blue colors represent droplets formed on CCN of 10/30/100 nm dry radius. The same data is shown in all panels. The left panel applies the difference between droplet radius

and critical radius on the vertical axis. The middle panel uses the ratio between the droplet radius and the critical radius. The right panel applies the difference between droplet radius and critical radius when the difference is positive (quarters Q1 and Q2), and the ratio between the droplet radius and the critical radius in Q3 and Q4 quarters.

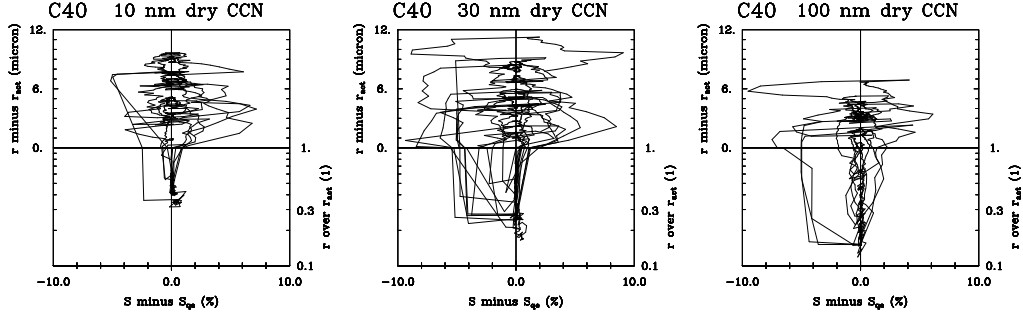

Figure 9. Representation of droplet trajectories on the generalized phase diagram for C40 simulation. Results are for 3 randomly selected droplets for (left panel) 10 nm, (middle panel) 30 nm, and (right panel) 100 nm dry CCN radius.

Figure 9 shows evolutions of several droplets using the generalized phase space diagram from the right panel of Fig. 8. The left, middle, and right panels show evolutions of droplet trajectories from additional 5 minute C40 simulations for three droplets

formed on 10, 30, and 100 nm dry CCN radius, respectively. (The three droplets for 10 nm dry CCN have already been used in Fig. 7). The generalized phase space diagram allows direct comparison of droplet trajectories that is independent of the CCN dry radius. Fig. 9 shows that larger dry CCN feature smaller $r$ and $r_{act}$ ratios for haze droplets. Moreover, the results in Fig. 9





seem to agree with those in Table 1, that is, there are fewer trajectories passing through the activation, and more passages between growth and evaporation, especially for haze droplets. However, one has to keep in mind that Fig. 9 only shows

randomly selected 9 droplets (three for each dry CCN radius), whereas Table 1 includes statistics for all droplets.

## 6 Summary

This paper introduces a phase space diagram that documents in a simple way microphysical processes leading to cloud droplet formation and growth. The diagram shown in Fig. 2 plots the haze or cloud droplet radius versus the difference between

ambient supersaturation $S$ and the equilibrium supersaturation $S_{eq}$ corresponding to the droplet radius. The difference combines droplet and environmental characteristics and it determines whether the droplet grows or evaporates. A similar diagram was used by Arabas and Shima (2017, see Fig. 1 there) to discuss CCN activation and subsequent deactivation of a droplet with a radius larger than the critical radius when the ambient supersaturation $S$ falls below the equilibrium supersaturation $S_{eq}$, that is, when $S - S_{eq}$ becomes negative. See also relevant discussions in Abade et al. (2018; see Fig. 2 there) and in Grabowski and

Pawlowska (2023). The current paper applies the diagram to depict the CCN activation, droplet growth, evaporation, and deactivation, especially relevant to microphysical processes taking place in fluctuating supersaturation due to cloud turbulence. For illustration, the diagram is applied to numerical simulations of the impact of turbulence on cloud base CCN activation and to cycles of CCN activation and deactivation inside the Pi cloud chamber. Cloud base activation is driven by adiabatic cooling of the rising air with turbulence modifying the sharp separation between activated and nonactivated CCN. Cloud chamber

simulations mimic microphysical transformations in the moist Rayleigh-Benard convection that are driven by turbulent mixing between humid warm and cold air. The Pi chamber supersaturation comes from the nonlinear dependence of the saturated water vapor pressure on temperature.

The cloud base CCN activation in fluctuating supersaturation due to cloud turbulence is discussed in Grabowski et al. (2022).

Here, we use one of the simulations to illustrate application of the phase diagram to depict phase space trajectories of CCN that: i) form cloud droplets that continuously grow above the cloud base; ii) CCN that activate and subsequently deactivate; and iii) CCN that never activate. For the cloud base activation, the partitioning between the three pathways i) to iii) depends on the dry CCN radius, with small CCN never getting activated and large CCN always activating to form cloud droplets. The new diagram is also applied to the microphysical processes within the Pi chamber discussed in Grabowski et al. (2024). We

use one of the Pi chamber simulations to document cycling of the CCN through activation, growth of cloud droplets, their evaporation, and CCN deactivation. The three dry CCN sizes selected from two CCN distributions considered in Grabowski et al. (2024), dry radius of 10, 30, and 100 nm radius, seem to move over the phase diagram in a similar fashion as documented in the Table 1. This is in contrast to the cloud base activation discussed in Grabowski et al. (2022).

A generalized form of the diagram is introduced in section 5. The motivation is to create a diagram that is independent of the dry CCN radius. The generalized phase diagram uses the difference between the droplet radius and the activation (critical)



radius for cloud droplets, and the ratio between the two for haze droplets. A linear scale is used for cloud droplets, and a log scale is used for haze droplets. For illustration, the generalized diagram is applied to Pi chamber simulations to compare activation, growth, evaporation, and deactivation of various dry CCN radii.


Application of the new diagram to simulations of entrainment and mixing for cumulus or stratocumulus clouds might be especially enlightening. This is not only because of the activation of entrained CCN, but also because of droplet evaporation and CCN deactivation as a result of cloud dilution. Detailed analysis of microphysical processes associated with cloud entrainment and mixing is possible applying the new phase space representation. We plan to pursue this line of research in the

near future.

**Data availability** Simulation data used in the analysis are available at https://dashrepo.ucar.edu/dataset/215_grabow.html (Grabowski 2021) and at https://gdex.ucar.edu/dataset/451.html (Grabowski 2024).

**Author contributions** WWG and HP formulated details of the diagram. WWG performed analysis of existing simulation results and drafted the figures. WWG and HP jointly wrote the paper.

**Competing interests** The authors declare that they have no conflict of interest.

*Acknowledgments*. This material is based upon work supported by the NSF National Center for Atmospheric Research, which is a major facility sponsored by the NSF under Cooperative Agreement 1852977. Comments on the early draft of this manuscript by NSF NCAR's Hugh Morrison are acknowledged.

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
