# Peer review of "Technical note: Phase space depiction of CCN activation and cloud droplet diffusional growth"

_EGUsphere, 2024_

## Referee Comment (RC1)

**Review of "Technical note: Phase space depiction of CCN activation and cloud droplet diffusional growth" by Grabowski and Pawlowska (egusphere-2024-4104)**

This manuscript introduces a phase space that helps to understand the activation, deactivation, condensation, and evaporation of haze particles and cloud droplets in a unified fashion. The phase space is applied to the results of two simulation cases, a turbulent rising parcel and a convection cloud chamber. Overall, this manuscript addresses an interesting and relevant topic. I have reviewed a previous version of this manuscript submitted to the Journal of the Atmospheric Sciences. The most important change from that version is its framing as a Technical Note, which I consider very appropriate as the manuscript does not contain substantial new findings but provides a concept to be applied in future studies. I consider this manuscript adequate for publication in Atmospheric Chemistry and Physics once my comments are addressed.

**Major Comments**

*Does it make sense to distinguish Q3 and Q4 for all radii smaller than the activation radius?* Conceptually, the regions Q3 and Q4 of the phase space are distinct, with the prior representing evaporation and the latter condensation. This is probably adequate for (super-)saturated conditions that a particle experiences inside the cloud. Outside the cloud, haze particles are usually in equilibrium with their environment, which is sustained by quick changes between evaporation and condensation (time series in Fig. 7). Thus, I recommend to introduce a fifth region to consider this equilibrium state. It should cover the entire range of $S - S_{eq}$ values, and all radii up to the equilibrium radius at saturation, $r_{eq} = \left( \kappa\, r_d^3 / A \right)^{1/2}$, which is obtained from equating (1) to 0.

**Minor Comments**

Sec. 1: I enjoyed reading this introduction to cloud droplet formation. However, I was wondering why the authors did not include the diffusional growth equation (and maybe an equation for the development of supersaturation in an adiabatic parcel). This would naturally integrate some of the dynamics considered in the introduced phase space.

Ll. 87 – 89: References to Nenes et al. (2001) and Mordy (1959) seem to be appropriate.

Ll. 241 – 243: What exactly is "not well visible in the right panel"?

L. 398: In the abstract (ll. 18 – 20), the authors promised to use the phase space to identify differences in droplet formation in the analyzed cases. This line seems to be the only location where this is actually done. Could the authors comment a little more on the differences of droplet formation in "natural and laboratory clouds"?

**Technical Comments**

Ll. 32 ff.: Change "Koehler" to "Köhler".

Ll. 43 ff.: Change "paper" to "technical note".

Ll. 250 – 258: Is "GYK24" different from "GKY24"?

**References**

Mordy, W. (1959). Computations of the growth by condensation of a population of cloud droplets. *Tellus*, *11*(1), 16-44.

Nenes, A., Ghan, S., Abdul-Razzak, H., Chuang, P. Y., & Seinfeld, J. H. (2001). Kinetic limitations on cloud droplet formation and impact on cloud albedo. *Tellus B: Chemical and Physical Meteorology*, *53*(2), 133-149.

---

## Author Comment (AC1)

Responses to the Reviewer 1 comments
(comments in black, responses in red)

This manuscript introduces a phase space that helps to understand the activation, deactivation, condensation, and evaporation of haze particles and cloud droplets in a unified fashion. The phase space is applied to the results of two simulation cases, a turbulent rising parcel and a convection cloud chamber. Overall, this manuscript addresses an interesting and relevant topic. I have reviewed a previous version of this manuscript submitted to the Journal of the Atmospheric Sciences. The most important change from that version is its framing as a Technical Note, which I consider very appropriate as the manuscript does not contain substantial new findings but provides a concept to be applied in future studies. I consider this manuscript adequate for publication in Atmospheric Chemistry and Physics once my comments are addressed.

We appreciate the sincere evaluation of our submission.

**Major Comments**

*Does it make sense to distinguish Q3 and Q4 for all radii smaller than the activation radius?*

Conceptually, the regions Q3 and Q4 of the phase space are distinct, with the prior representing evaporation and the latter condensation. This is probably adequate for (super-)saturated conditions that a particle experiences inside the cloud. Outside the cloud, haze particles are usually in equilibrium with their environment, which is sustained by quick changes between evaporation and condensation (time series in Fig. 7). Thus, I recommend to introduce a fifth region to consider this equilibrium state. It should cover the entire range of $S - S_{eq}$ values, and all radii up to the equilibrium radius at saturation.

On the Koehler curve, $S_{eq} = 0$ points (i.e., those with radii equal to the critical radii divided by square root of 3) correspond to the equilibrium droplet radii for vanishing environmental supersaturation (i.e., RH = 100%). We added those points to Fig. 1. The only significance of the wet radius corresponding to RH = 100% is that a haze droplet outside a cloud (i.e., with RH < 100%) has the equilibrium radius smaller than its $S_{eq} = 0$ radius. We mention that in the discussion of Fig. 1.

On our diagram, the "equilibrium region" that the reviewer suggests is a 1D space along the $S - S_{eq} = 0$ line and radii smaller that the critical radius divided by square root of 3. Those are haze droplets that have equilibrium radii for RH < 100%, that is, haze droplets outside the cloud. In contrast to the critical radius, we do not see any significance of growing droplets passing this radius. For illustration, we show below Fig. 5 from the manuscript with extra lines in both panels that correspond to the critical radius divided by the square root of 3. Do those lines bring additional information into the panels? We do not think so.

[Figure]

[Figure]

Figure 5 from the manuscript with extra lines in left and right panels showing the radius corresponding to the zero of the Koehler curve. Please compare with the figure 5 in the text that has no additional lines.

**Minor Comments**

Sec. 1: I enjoyed reading this introduction to cloud droplet formation. However, I was wondering why the authors did not include the diffusional growth equation (and maybe an equation for the development of supersaturation in an adiabatic parcel). This would naturally integrate some of the dynamics considered in the introduced phase space.

Per the suggestion, we revised the introduction and added a droplet growth equation. Since the second case we consider, the Pi chamber case, is different than the adiabatic parcel, we do not feel bringing adiabatic parcel equations is needed.

Ll. 87 – 89: References to Nenes et al. (2001) and Mordy (1959) seem to be appropriate.

In the revised introduction, we include references to Mordy and Nenes et al .

Ll. 241 – 243: What exactly is "not well visible in the right panel"?

Green lines are congested and thus not well visible. We removed the statement in question.

L. 398: In the abstract (ll. 18 – 20), the authors promised to use the phase space to identify differences in droplet formation in the analyzed cases. This line seems to be the only location where this is actually done. Could the authors comment a little more on the differences of droplet formation in "natural and laboratory clouds"?

We feel the discussion in sections 3 and 4 are exactly what the reviewer is asking about. We added a sentence in the summary section that highlights the differences.

**Technical Comments**

Ll. 32 ff.: Change "Koehler" to "Köhler".

Done.

Ll. 43 ff.: Change "paper" to "technical note".

Done.

Ll. 250 – 258: Is "GYK24" different from "GKY24"?

GKY24 is the correct abbreviation. This error has been corrected.

---

## Author Comment (AC2)

Responses to the Reviewer 2 comments
(comments in black, responses in red)

This technical note introduces a visualization framework for analyzing aerosol/cloud particle trajectories in thermodynamic/water phase space, with a specific application to outputs from dynamical models that employ Lagrangian particle-based approaches. Examples from two different environments (idealized atmospheric cloud vs. laboratory chamber cloud) are given to show how differences in the driving dynamics map onto the phase diagram, allowing prospective adapters to see for themselves how the proposed approach differentiates dynamical regimes. This note is a valuable addition to the literature and I recommend it for publication after the following comments are addressed.

We appreciate the positive evaluation of our submission.

**Comments**

L82-83: Is it commonly known that "large CCN typically lag the environmental RH increase"? I.e., is this something that one would know after reading a canonical text like Pruppacher and Klett or Yau and Rogers? If not, could you provide a citation supporting this statement?

We do not think this is a common knowledge, but it is appreciated by those who study CCN deliquescence and activation. It has been mentioned in the introduction to Grabowski et al. (2022b) and illustrated by the simulations there. The part of the text the reviewer refers to has changed. Per Reviewer 1 suggestions, we added a discussion of that aspect and included references to Mordy (1959), Chuang et al. (1997), and Nenes et al. (2001). The revised text reads (Eq. 2 is the droplet growth equation added per Reviewer 1 suggestion):

"This can be shown by considering time scale characterizing droplet growth that can be taken as $r$ over $dr/dt$. Neglecting the impact of the equilibrium supersaturation $S_{eq}$ in (2), the droplet growth time scale is proportional to $r^2$ (see the derivation in Mordy 1959 that considers the impact of $S_{eq}$ on the droplet growth time scale). Chuang et al. (1997) puts the droplet growth time scale in the context of the time scale characterizing the environmental supersaturation change as, for instance, in the rising adiabatic parcel. Such considerations are further refined in Nenes et al. (2001) that discusses time-scale limitations that may or may not lead to eventual activation of a given CCN together with a possible deactivation of already activated cloud droplets."

Section 3, paragraph 1 (L153-163): I don't think you say explicitly in this model description paragraph that you're using an implementation of the super-droplet method. This would be helpful, even though it should already be obvious.

We added the following sentence:

"Lagrangian particle-based microphysics (Shima et al. 2009) is applied to represent deliquesced CCN and activated cloud droplets."

L227 and Fig. 5, left panel x-axis label: Why do you use droplet radius to describe particles that may or may not in fact be water droplets? Would it make more sense to use the general "particle radius?"

We do not understand this comment. All "particles" throughout the manuscript are water droplets, either haze droplets (i.e., deliquesced CCN) or cloud droplets (i.e., activated CCN).

L250, 256, 258: GKY24 became GYK24. Please correct.

Corrected. This was also spotted by the Reviewer 1.

L250: Following the setup of one of the laboratory experiments (add "the")

Added. Thanks.

---

## Author Response (AR3)

This manuscript introduces a phase space that helps to understand the activation, deactivation, condensation, and evaporation of haze particles and cloud droplets in a unified fashion. The phase space is applied to the results of two simulation cases, a turbulent rising parcel and a convection cloud chamber. Overall, this manuscript addresses an interesting and relevant topic. I have reviewed a previous version of this manuscript submitted to the Journal of the Atmospheric Sciences. The most important change from that version is its framing as a Technical Note, which I consider very appropriate as the manuscript does not contain substantial new findings but provides a concept to be applied in future studies. I consider this manuscript adequate for publication in Atmospheric Chemistry and Physics once my comments are addressed.

We appreciate the sincere evaluation of our submission.

**Major Comments**

*Does it make sense to distinguish Q3 and Q4 for all radii smaller than the activation radius?*

Conceptually, the regions Q3 and Q4 of the phase space are distinct, with the prior representing evaporation and the latter condensation. This is probably adequate for (super-)saturated conditions that a particle experiences inside the cloud. Outside the cloud, haze particles are usually in equilibrium with their environment, which is sustained by quick changes between evaporation and condensation (time series in Fig. 7). Thus, I recommend to introduce a fifth region to consider this equilibrium state. It should cover the entire range of $S - S_{eq}$ values, and all radii up to the equilibrium radius at saturation.

On the Koehler curve, $S_{eq} = 0$ points (i.e., those with radii equal to the critical radii divided by square root of 3) correspond to the equilibrium droplet radii for vanishing environmental supersaturation (i.e., RH = 100%). We added those points to Fig. 1. The only significance of the wet radius corresponding to RH = 100% is that a haze droplet outside a cloud (i.e., with RH < 100%) has the equilibrium radius smaller than its $S_{eq} = 0$ radius. We mention that in the discussion of Fig. 1.

On our diagram, the "equilibrium region" that the reviewer suggests is a 1D space along the $S - S_{eq} = 0$ line and radii smaller that the critical radius divided by square root of 3. Those are haze droplets that have equilibrium radii for RH < 100%, that is, haze droplets outside the cloud. In contrast to the critical radius, we do not see any significance of growing droplets passing this radius. For illustration, we show below Fig. 5 from the manuscript with extra lines in both panels that correspond to the critical radius divided by the square root of 3. Do those lines bring additional information into the panels? We do not think so.

[Figure]

[Figure]

Figure 5 from the manuscript with extra lines in left and right panels showing the radius corresponding to the zero of the Koehler curve. Please compare with the figure 5 in the text that has no additional lines.

**Minor Comments**

Sec. 1: I enjoyed reading this introduction to cloud droplet formation. However, I was wondering why the authors did not include the diffusional growth equation (and maybe an equation for the development of supersaturation in an adiabatic parcel). This would naturally integrate some of the dynamics considered in the introduced phase space.

Per the suggestion, we revised the introduction and added a droplet growth equation. Since the second case we consider, the Pi chamber case, is different than the adiabatic parcel, we do not feel bringing adiabatic parcel equations is needed.

Ll. 87 – 89: References to Nenes et al. (2001) and Mordy (1959) seem to be appropriate.

In the revised introduction, we include references to Mordy and Nenes et al .

Ll. 241 – 243: What exactly is "not well visible in the right panel"?

Green lines are congested and thus not well visible. We removed the statement in question.

L. 398: In the abstract (ll. 18 – 20), the authors promised to use the phase space to identify differences in droplet formation in the analyzed cases. This line seems to be the only location where this is actually done. Could the authors comment a little more on the differences of droplet formation in "natural and laboratory clouds"?

We feel the discussion in sections 3 and 4 are exactly what the reviewer is asking about. We added a sentence in the summary section that highlights the differences.

**Technical Comments**

Ll. 32 ff.: Change "Koehler" to "Köhler".

Done.

Ll. 43 ff.: Change "paper" to "technical note".

Done.

Ll. 250 – 258: Is "GYK24" different from "GKY24"?

GKY24 is the correct abbreviation. This error has been corrected.

Responses to the Reviewer 2 comments
(comments in black, responses in red)

This technical note introduces a visualization framework for analyzing aerosol/cloud particle trajectories in thermodynamic/water phase space, with a specific application to outputs from dynamical models that employ Lagrangian particle-based approaches. Examples from two different environments (idealized atmospheric cloud vs. laboratory chamber cloud) are given to show how differences in the driving dynamics map onto the phase diagram, allowing prospective adapters to see for themselves how the proposed approach differentiates dynamical regimes. This note is a valuable addition to the literature and I recommend it for publication after the following comments are addressed.

We appreciate the positive evaluation of our submission.

**Comments**

L82-83: Is it commonly known that "large CCN typically lag the environmental RH increase"? I.e., is this something that one would know after reading a canonical text like Pruppacher and Klett or Yau and Rogers? If not, could you provide a citation supporting this statement?

We do not think this is a common knowledge, but it is appreciated by those who study CCN deliquescence and activation. It has been mentioned in the introduction to Grabowski et al. (2022b) and illustrated by the simulations there. The part of the text the reviewer refers to has changed. Per Reviewer 1 suggestions, we added a discussion of that aspect and included references to Mordy (1959), Chuang et al. (1997), and Nenes et al. (2001). The revised text reads (Eq. 2 is the droplet growth equation added per Reviewer 1 suggestion):

"This can be shown by considering time scale characterizing droplet growth that can be taken as $r$ over $dr/dt$. Neglecting the impact of the equilibrium supersaturation $S_{eq}$ in (2), the droplet growth time scale is proportional to $r^2$ (see the derivation in Mordy 1959 that considers the impact of $S_{eq}$ on the droplet growth time scale). Chuang et al. (1997) puts the droplet growth time scale in the context of the time scale characterizing the environmental supersaturation change as, for instance, in the rising adiabatic parcel. Such considerations are further refined in Nenes et al. (2001) that discusses time-scale limitations that may or may not lead to eventual activation of a given CCN together with a possible deactivation of already activated cloud droplets."

Section 3, paragraph 1 (L153-163): I don't think you say explicitly in this model description paragraph that you're using an implementation of the super-droplet method. This would be helpful, even though it should already be obvious.

We added the following sentence:

"Lagrangian particle-based microphysics (Shima et al. 2009) is applied to represent deliquesced CCN and activated cloud droplets."

L227 and Fig. 5, left panel x-axis label: Why do you use droplet radius to describe particles that may or may not in fact be water droplets? Would it make more sense to use the general "particle radius?"

We do not understand this comment. All "particles" throughout the manuscript are water droplets, either haze droplets (i.e., deliquesced CCN) or cloud droplets (i.e., activated CCN).

L250, 256, 258: GKY24 became GYK24. Please correct.

Corrected. This was also spotted by the Reviewer 1.

L250: Following the setup of one of the laboratory experiments (add "the")

Added. Thanks.

Responses to the Editor comments
(comments in black, responses in red)

I thank the authors for the generally comprehensive responses to reviewer comments.

In the instance where a reviewer became confused about the usage of "droplet" to refer to unactivated aerosol and no revision was yet made: I can understand their confusion and agree that "particle" may be in wider use to span aerosol and hydrometeor categories (e.g., Jacobson,

2005). To avoid similar confusion for other readers, and since the terms "aerosol", "CCN", and "particle" are all used as well, please clarify how you will be using the term "droplet" at first use in that dual capacity. I believe that may be where the Köhler equation is introduced. A possible clarification could be as follows: "We hereafter refer to the spherical particle as a droplet, regardless of whether it is at equilibrium with water vapor (as a CCN) or activated (as a cloud droplet)." or something of that nature. Thank you in advance for making that clarification. Such terminology confusion could be an indicator of education primarily in the aerosol or cloud community.

Thanks for the comment and suggestion. We prefer not to use the word "particle" when referring to liquid droplets. Right now, the word "particle" is only used in in the very beginning of the introduction ("aerosol particles" in line 29) and throughout the manuscript in "particle-based microphysics" (because this approach is also used for ice). As stated in our response to the Reviewer 2, all particles in our manuscript are water droplets, either deliquesced CCN prior to activation, or activated CCN. Referring to them as "particles" would be confusing in our view. As a compromise, we added a footnote after the sentence "The equilibrium water vapor pressure over a spherical droplet containing a soluble material is described by the Köhler equation." The footnote reads: "Throughout the manuscript, water droplets include both deliquesced CCN and activated CCN. The former are referred to as the haze droplets and the latter as the cloud droplets." Such an addition resulted in small changes throughout the text to ensure that haze droplet and cloud droplet terms are used according to the footnote.

On a second reading of the brief review of relevant physics, I suggest to include a reference regarding homogeneous nucleation for the interested reader. With such a reference, a more precise statement could be "In an Earth atmosphere without aerosol particles, droplet formation rates would remain vanishingly weak (e.g., Wyslouzil and Wölk, 2016)." I defer, but suggest that readers interested in that underlying physics could appreciate to see a reference for further reading.

I think the Editor comment concerns the sentence: "Without CCN, it is impossible to form a water droplet directly from water vapor in the Earth atmosphere." We changed "in the Earth atmosphere" into "in the Earth natural clouds" (which we believe is a correct statement) and added a reference suggested by the Editor. The revised sentence reads:

"Without CCN, it is impossible to form a water droplet directly from water vapor in the Earth natural clouds (see Wyslouzil and Wölk 2016 for a review of theoretical and experimental studies of the homogeneous water droplet formation)."